# Real-World Outcomes and Treatments Patterns Prior and after the Introduction of First-Line Immunotherapy for the Treatment of Metastatic Non-Small Cell Lung Cancer

**DOI:** 10.3390/cancers14184481

**Published:** 2022-09-15

**Authors:** Valentina Danesi, Ilaria Massa, Flavia Foca, Angelo Delmonte, Lucio Crinò, Giuseppe Bronte, Maria Ragonesi, Roberta Maltoni, Silvia Manunta, Paola Cravero, Kalliopi Andrikou, Ilaria Priano, William Balzi, Nicola Gentili, Thomas Burke, Mattia Altini

**Affiliations:** 1Outcome Research, Healthcare Administration, IRCCS Istituto Romagnolo per lo Studio dei Tumori (IRST) “Dino Amadori”, 47014 Meldola, Italy; 2Unit of Biostatistics and Clinical Trials, IRCCS Istituto Romagnolo per lo Studio dei Tumori (IRST) “Dino Amadori”, 47014 Meldola, Italy; 3Department of Medical Oncology, IRCCS Istituto Romagnolo per lo Studio dei Tumori (IRST) “Dino Amadori”, 47014 Meldola, Italy; 4Nursing Service, IRCCS Istituto Romagnolo per lo Studio dei Tumori (IRST) “Dino Amadori”, 47014 Meldola, Italy; 5AULSS5, UOC Oncologia, Ospedale Santa Maria della Misericordia, 45100 Rovigo, Italy; 6Center for Observational and Real World Evidence, Merck & Co Inc., Kenilworth, NJ 07033, USA; 7Healthcare Administration, Azienda Unità Sanitaria Locale della Romagna, 48121 Ravenna, Italy

**Keywords:** non-small cell lung cancers (NSCLC), real-world evidence, observational study, overall survival (OS), real-world progression-free survival (rwPFS), immune checkpoint inhibitors (ICIs), first-line treatment, cancer immunotherapy

## Abstract

**Simple Summary:**

The advent of immuno-oncology (IO) agents, particularly immune checkpoint inhibitors (ICIs), has changed the treatment landscape of non-small cell lung cancer (NSCLC). We performed a retro-prospective study to describe the patients’ outcomes prior to and after the local regulatory approval of pembrolizumab as a first-line (1L) treatment in the real-world setting of an Italian cancer centre. Analyses were performed of a total of 694 patients with no or unknown oncogene addicted tumour, grouped into Pre- (*n* = 344) and Post- (*n* = 350) 1L IO populations. The study provides evidence of improvements in overall survival associated with the introduction of 1L immunotherapy, suggesting that receiving immunotherapy in the first-line rather than in the second- or later lines of treatment may be more favourable.

**Abstract:**

Background: This study provides insights into the treatment use and outcomes of metastatic non-small cell lung cancer (NSCLC) patients in a real-world setting prior to and after the availability of immuno-oncology (IO) regimens in the first line (1L). Methods: Metastatic NSCLC patients, who initiated systemic 1L anticancer treatment from 2014 to 2020, were identified from health records. Patients were grouped into Pre-1L IO and Post-1L IO, according to the availability of pembrolizumab 1L monotherapy at the date of initiating 1L systemic anticancer treatment. Patient characteristics, treatment patterns and outcomes were assessed by the cohort. Overall survival (OS) and real-world progression-free survival (rwPFS) were calculated using the Kaplan-Meier method. Results: The most common 1L treatment was platinum-based chemotherapy regimens in both groups (≥46%), followed by single-agent chemotherapy (27.0%) in Pre-1L IO and pembrolizumab (26.0%) in Post-1L IO. Median OS was 6.2 (95% CI 5.5–7.4) in Pre- and 8.9 months (95% CI 7.5–10.6) in Post-1L IO, while rwPFS was 3.7 (95% CI 3.3–4.2) and 4.7 months (95% CI 3.9–5.7), respectively. Conclusions: Even if a small proportion of patients received a 1L IO, the data showed an improved survival outcomes in the Post-1L IO group.

## 1. Introduction

Before the approval of first-line (1L) immuno-oncology (IO) agents for non-oncogene-addicted metastatic non-small cell lung cancer (mNSCLC), platinum doublet chemotherapy with or without bevacizumab or single-agent chemotherapy were the standard of care for treatment of patients with a good performance status (PS) (Eastern Cooperative Oncology group (ECOG) 0 or 1) or ECOG PS = 2, respectively. However, the efficacy of these conventional treatments was modest, as chemotherapy is associated with a relatively low and non-durable response rate and limited survival benefits [1,2,3]. On the other hand, targeted therapies were limited to a small subset of molecularly selected patients.

The therapeutic landscape in metastatic NSCLC changed with the approval of immunotherapy agents, particularly immune checkpoint inhibitors (ICIs), targeting the programmed death-1 (PD-1)/programmed death-ligand 1 (PD-L1) pathway for both treatment-näive and previously treated disease, irrespective of histology. ICIs became part of the standard of care for lung cancer patients, demonstrating an overall survival benefit compared to chemotherapy [4,5,6,7]. In April 2016, nivolumab was approved for NSCLC, receiving market access approval from the Italian Medicines Agency (AIFA) at the national level for the second-line (2L) treatment of NSCLC patients [8]. Later, immunotherapy moved into the 1L setting: pembrolizumab received market access approval from AIFA at the national level in May 2017 for both 1L metastatic NSCLC (monotherapy for metastatic NSCLC with PD-L1 ≥50% and without EGFR- or ALK-positive tumor mutations) and 2L treatment (monotherapy for locally advanced or metastatic NSCLC with PD-L1 ≥1% who have received at least one prior chemotherapy) [9]. The regional access of nivolumab and pembrolizumab monotherapy for NSCLC was possible from February 2017 and July 2017, respectively [10], while atezolizumab for NSCLC was approved at the regional level in August 2018 [10]. Likewise, the positive results from clinical trials led to the approval of pembrolizumab in combination with pemetrexed and platinum chemotherapy as a 1L therapy in patients with non-squamous NSCLC in adults whose tumors had PD-L1 <50% and who had developed no EGFR- or ALK-positive mutations since January 2020 in the Emilia-Romagna region.

Although clinical trials are the gold standard to evaluate the efficacy and safety of treatment regimens, the selected patient sample is not representative of the entire patient population treated in clinical practice [11]. Real-world analyses provide a more comprehensive picture by documenting the treatment patterns and survival outcomes of patients treated in clinical practice. This may help clinicians to make better treatment decisions. Furthermore, the expected changes in the outcomes of mNSCLC patients after the clinical implementation of 1L pembrolizumab monotherapy have not been largely investigated in real-world cohorts.

In light of the introduction of 1L IO regimens for the treatment of mNSCLC, we conducted a non-interventional study to generate real-life practice data among non-oncogene-addicted mNSCLC patients, treated at an Italian hospital. The focus of the study is to describe the patients’ outcomes prior to and after the local regulatory approval of 1L pembrolizumab to complement the growing real-world literature on ICIs in the treatment of NSCLC patients. 

## 2. Methods

### 2.1. Study Design and Data Source

This was a single-center, retro-prospective observational study, conducted in IRCCS Istituto Romagnolo per lo Studio dei Tumori (IRST) “Dino Amadori” of Meldola (Forlì-Cesena), Italy. The study population and clinical dataset were extracted from Electronic Health Records (her) of IRST, and maintained during routine clinical practice, including information on demographics, clinical, tumour assessment, molecular characteristics and the administered treatment(s). Data collection was supplemented by a manual review of unstructured data (i.e., clinical notes, radiology reports or pathology reports). The study involved multiple data extractions (retrospective and prospective) from her; the last extraction took place in April 2021. Each data extraction was followed by data cleaning and data quality assessments, focusing on key variables including patient demographics, disease characteristics and cancer treatments. The initiation of a 1L systemic anticancer therapy for the treatment of mNSCLC was defined as an index date. Patients were followed from this date until death, the last documented follow-up, or the end of the study period (31 December 2020), whichever occurred first. This ensured at least 6 months of potential follow-up time for patients in the Post-1L IO cohort. Mortality information was obtained and verified from the mortality register (ReM) database of the Emilia-Romagna region. Informed consent forms (ICF) were collected for living patients. Living patients without a signed ICF were excluded. This study was approved by the Scientific and Medical Committee and the Ethic Committee of IRST-IRCCS Area Vasta Romagna (CEROM). 

### 2.2. Study Population

Eligible patients were: (i) aged ≥18 years; (ii) with a confirmed diagnosis of NSCLC presenting with stage IV or stage IIIB with rapidly progressive (IIIBrp), who experienced disease progression to stage IV within 6 months from the first anticancer treatment, without completing induction therapy, both radiotherapy and chemotherapy; (iii) initiated 1L treatment between 1 January 2014 and 30 June 2020 at IRST; (iv) patients enrolled after signing the ICF or death. Patients enrolled in an interventional clinical trial during the study period were included, with the blinding of information regarding the interventional clinical trial details. For the purpose of this paper, patients with a known oncogene-addicted NSCLC as epidermal growth factor receptor (EGFR) or an anaplastic lymphoma kinase (ALK) or receptor tyrosine kinase (ROS1) gene alteration (translocation, fusion, amplification) were excluded from the analysis. 

### 2.3. Cohort Description

The patients with no or an unknown oncogene-addicted tumour (EGFR, ALK and ROS1 negative or unknown) were separated into two groups based on the availability of the first ICI available as 1L therapy in the Emilia-Romagna region (pembrolizumab in PD-L1 tumour proportion score (TPS) ≥50% mNSCLC):

(1) Pre-1L IO group: eligible patients who started 1L treatment from January 2014 to June 2017 before the first-line ICI was available in the Emilia-Romagna region; 

(2) Post-1L IO group: included eligible patients who started 1L treatment from July 2017 to June 2020, after first-line ICI was available in the Emilia-Romagna region.

For the assessment of 1L and 2L treatment patterns, the Post-1L IO population was further grouped according to:

(1) Patients with PD-L1 expression on at least 50% of tumor cells (i.e., TPS ≥ 50%);

(2) Patients with PD-L1 expression on less than 50% of tumor cells (i.e., TPS < 50% or null).

### 2.4. Assessments and Study Endpoints

The line and duration of therapy were identified using EHR and a rule-based algorithm. By definition, maintenance therapy, interruptions, or the replacement of one drug in the combination regimen with another due to toxicity (e.g., the replacement of carboplatin with cisplatin) did not advance the line of therapy. The length of each line was estimated to reach from the start of the treatment to the earliest occurrence among: (a) the start of a new agent not included in the previous therapy regimen; (b) a gap of >42 days in drug administration; (c) death. 

The primary outcomes that were analyzed included overall survival (OS) and real-world progression-free survival (rwPFS). OS was defined as the period from the index date to death due to any cause. All patients verified to be alive as of the last data export (April 2021) were censored for death asthe end of follow-up (31 December 2020).

Conversely, for patients whose vital status could not be verified, the survival time was censored as the date of the last visit or last activity before the end of follow-up (31 December 2020).

rwPFS was defined as the time from the first dose of 1L treatment to documented clinical disease progression, the initiation of a new line of therapy, or death from any cause, whichever occurred first. For patients treated with ICIs, in case of radiological progression without a clinical worsening and with a continuation of the same treatment, this was not considered an event, and the date of progression was based on ICI treatment discontinuation [12]. Patients without an event were censored at the date of the last clinical tumour assessment. Patients with a progressive event occurring within 14 days of the start of 1L therapy were excluded from rwPFS analysis, as well as patients who did not even undergo a tumour assessment within the study period. 

The real-world tumour response rate (rwORR) was defined as the proportion of patients with a radiologically documented or clinically-assessed complete response (CR) or partial response (PR). Real-world disease control rate was defined as proportion of patients with radiologically documented or clinician-assessed complete response or partial response or stable disease (SD). Outcomes analyses were conducted separately for the Pre- and Post-IO group.

### 2.5. Statistical Analysis 

Due to the descriptive nature of the study and lack of hypothesis testing, a formal calculation of sample size and statistical power was not performed. Descriptive tables were used to summarize baseline and treatment characteristics. Continuous variables were presented as median (min–max values) and categorical variables were presented as absolute and relative frequencies. Time-to-event data (OS, rwPFS) were described using Kaplan-Meier curves. Ninety-five percent of the confidence intervals (95% CI) were calculated using non-parametric methods. The rate of objective response (rwORR) and the disease control rate (rwDCR) were calculated with an exact 95% CI using standard methods based on binomial distribution. Statistical analyses were carried out using STATA/MP 15.0 for Windows (Stata Corp LP, College Station, TX, USA). A Sankey diagram was generated to show patter across treatment lines using the htmltools, htmlwidgets and networD3 packages in R software (version 4.1.0; The R Project for statistical Computing; http://www.r-project.org).

## 3. Results

### 3.1. Patients

Among the 1002 patients with a confirmed diagnosis of IIIBrp/IV stage NSCLC between January 2014 and June 2020, a total of 846 patients met the eligibility criteria (Figure 1). Among the overall patients, only 20 patients (2.4%) were enrolled with stage IIIBrp; the remaining 826 patients were enrolled with stage IV. Among the overall sample (*n* = 846), 270 patients (31.9%) were tested for EGFR mutations, and ALK and ROS1 translocations. The proportion of patients who were tested for one or two of these biomarkers was 46.9% (*n* = 396). Conversely, a total of 180 (21.2%) patients were not tested or had unknown status. Oncogene-addicted mutations were detected in 152 patients (17.8%), while the remaining 694 patients were considered for the analysis of this study. Of those, 344 (49.6%) were found in the Pre-1L IO group, while 350 (50.4%) were found in the Post-1L IO group. 

In the Post-1L IO (*n* = 350) population with a known tumour PD-L1 expression status, a total of 84 (24.0%) and 110 (31.4%) patients had a tumour proportion score (TPS) ≥50% and 1% ≤ TPS ≤ 49%, respectively. A total of 114 patients had TPS < 1% (Figure 1). The number of patients not tested for PD-L1 biomarkers was 42 (12.0%) and included patients without sufficient or adequate material for the assessment and patients for whom testing was not conducted or planned.

For each cohort, the patient demographic and clinical characteristics are summarized in Table 1. In both cohorts, the majority of patients were under 70 years old, with a median age of 68.5 (range: 38.4–90.1) in Pre-1L IO cohort and 71.5 (range: 39.9–89.9) in Post-1L IO cohort. There was a similar proportion of female patients in Pre- and Post-1L IO cohort (34.3% vs. 32.0%). In both groups, the majority had a history of smoking (≥91.8%). The distribution of histological types was similar across the two cohorts. The predominant histology was adenocarcinoma, with similar proportion (76.2% in Pre- and 77.8% in Post-1L IO groups), whereas squamous cell carcinoma was 16.8% and 20.5%, respectively. A total of 85.1% patients in the Pre-1L and 81.9% in Post-1L IO had an ECOG PS of 0–1 at IIIBrp/IV stage diagnosis. Conversely, 14.9% and 18.1% (Pre- and Post-1L IO, respectively) of patients had ECOG performance status ≥2. Among the known metastatic sites, contralateral lung was the most common site of metastasis in both cohorts, representing more than one-third of patients.

### 3.2. First-Line Treatment Patterns 

The first-line treatment patterns are summarized in Table 2. In the Pre-1L IO cohort, 224 patients (65.1%) received multi-agent chemotherapy regimens (mainly pemetrexed +/−platinum), while 93 patients (27.0%) received single-agent chemotherapy regimens, and 25 patients (7.3%) were included in clinical trials. In the Post-1L IO cohort, multi- (58.2%) and single-agent (27.7%) chemotherapy were the most common regimens in patients with PD-L1 expression <50%. Carboplatinum + gemcitabine was the most representative treatment (28.3%) among patients treated with multi-agent chemotherapy. Conversely, among patients with TPS ≥50%, the most common 1L regimen was pembrolizumab, administered in 71 patients (84.5%). A total of 43.9% and 37.7% patients continued to receive 2L treatment in the Pre- and Post-1L IO cohort, respectively. The sequence most utilized in Pre-1L IO was multi-agent chemotherapy as the 1L, followed by clinical trial (Figure 2), and multi-agents chemotherapy, followed by a PD-1/PD-L1 inhibitor single agent in Post-1L IO (Figure 2).

Details on the subsequent lines of 2L and 3L therapy are reported in Appendix A. 

### 3.3. Patients’ Outcomes 

Median observed patient follow-up was 62.7 months (range: 26.6–78.9) and 25.3 (range: 6.1–41.1) months in Pre- and Post-1L IO, respectively. 

About 95.6% (*n* = 329) of patients in the Pre- and 72.3% (*n* = 253) in the Post-1L IO cohort died within the observation period. A total of 16 patients, 3 (0.9%) in the Pre- and 13 (3.7%) in the Post-IO cohort were lost to follow-up. The median time from IIIBrp/IV stage diagnosis to start of first-line therapy was 41.1 (range: 0–279) days. 

As shown in Figure 3a, the median OS time from the initiation of 1L therapy was 6.2 months (95% CI 5.5–7.4) in the Pre-1L IO cohort, with an estimated survival rate of 51.2% and 28.2% at 6 and 12 months, respectively. In the Post-1L IO, the median OS was 8.9 months (95% CI 7.5–10.6), and the estimated percentage of patients who were alive at 6 and 12 months was 59.1% and 42.5%, respectively. 

The exclusion criteria for rwPFS were as follows: 15 and 14 patients were excluded from Pre- and Post-1L IO analysis of rwPFS, respectively. In the Pre-1L IO cohort, the median rwPFS associated with the 1L was 3.7 months (95% CI 3.3–4.2 months), and the percentage of patients without an event at 6 and 12 months was 29.5% and 9.1%, respectively (Figure 3b). The median rwPFS in the Post-1L IO cohort was 4.7 months (95% CI 3.9–5.7 months) and the percentage of patients without an event at 6 and 12 months was 42.9% and 23.9%, respectively. 

The greatest survival improvements were observed in Post-1L IO patients receiving first-line ICI with a median OS of 15.5 months (95% CI 9.8–23.5 months) and a median rwPFS of 12.1 months (95% CI 6.8–15.1 months) (Figure 3c,d). Further outcome data are reported in Appendix A. 

### 3.4. Response to the First-Line Treatment 

Partial responses were observed in 51 (24.4%) and 83 (29.7%) in Pre- and Post-1L IO cohort, respectively, whereas complete responses were observed in 1 (0.5%) and 13 (4.7%), respectively. In Pre- and Post-1L IO cohorts, the disease control rate was 55.8% (95% CI: 49.3–62.2) and 59.0% (95% CI: 52.9–64.9), respectively. For further details, see Appendix A.

## 4. Discussion

This retro-prospective observational study provides insight into the current treatment use and outcomes of metastatic NSCLC patients in a real-world setting prior to and after the availability of immune regimens in 1L. Our findings show that the use of traditional chemotherapy alone is gradually being overtaken by the introduction of immunotherapy, which is improving survival outcomes in a real-world population. Patients who receive immunotherapy will certainly receive promising outcomes in the future; however, additional evaluations need to improve IO’s efficacy by developing robust predictive biomarkers to better select patients and understanding which drugs or combination of drugs might provide the strongest benefit.

The Keynote-001 NSCLC expansion cohorts demonstrated that PD-L1’s expression in at least 50% of tumor cells correlated with the improved efficacy of pembrolizumab as a monotherapy in treatment-naïve and previously treated advanced NSCLC [13]. Subsequently, three phase-III clinical trials demonstrated the benefit of pembrolizumab monotherapy in PD-L1 expressing previously treated advanced NSCLC in Keynote-010 [14] and treatment-naïve PD-L1 expressing advanced/metastatic NSCLC [4,15]. Recently, the addition of pembrolizumab to platinum-based chemotherapy was used in treatment-naïve advanced NSCLC patients with a PD-L1 expression of ≥1% [16,17] Similarly, other IOs have demonstrated a clear improvement in outcomes as a monotherapy or in combination of ICIs with chemotherapy versus chemotherapy alone, regardless of PD-L1 expression levels [16,17,18,19]. On the basis of these trials, the use of 1L IO has been increasing since FDA approval was first received in 2016, based on KEYNOTE-024 [4]. 

This observational study provides an opportunity to evaluate the real-world clinical management of patients who initiated systemic therapy for mNSCLC at IRST in Italy from 2014 to 2020, providing a comprehensive evaluation of outcomes prior to and after the availability of immune regimens as a 1L treatment for mNSCLC patients. We focused on real-world systemic anticancer therapy utilization, OS and rwPFS outcomes for patients diagnosed with stage IIIBrp/IV NSCLC with EGFR, ALK and ROS1 statuses of negative or unknown, grouping the overall population into two cohorts, based on 1L treatment started prior to or after pembrolizumab became the standard of care in the Emilia-Romagna region. As would be expected from a real-world cohort, some of the characteristics of our population differed from clinical trials. Our population included many older patients and some patients had an ECOG performance status of 2–3. As expected, patients in Pre- and Post-1L IO cohorts have similar demographic characteristics: the majority of the mNSCLC population was male, under 70 years of age, had a smoking history and showed an adenocarcinoma histology. 

Survival estimates were generally shorter than those reported in previous real-world studies for both cohorts [20,21,22,23,24,25]. There are possibly some underlying EGFR/ALK/ROS1-positive patients, as not all patients were tested for underlying mutations/aberrations. In our opinion, this variability could be explained, considering that most real-word studies include patients irrespective of EGFR, ALK and ROS1 status, who received target therapy, and others mainly focused on ICIs treatment were the most-used therapy in this sample. Moreover, the time between stage IV diagnosis (referred to the detection of metastasis lesions) and the start of treatment differed from one hospital to another. The organizational characteristics of a hospital regarding the management of patients with lung cancer may influence the timeliness of care. The rapid-access program for lung cancer care, coordination and communication across multiple discipline (e.g., radiology, pulmonology, medical oncology, surgery), as well as a multidisciplinary approach and molecular testing strategy, may be critical and influence diagnostic and treatment delay [26,27]. However, 1L IO was only well-represented in those with PD-L1 ≥50%, representing 20% of the Post-1L IO group. Similar results were described in a recently published study [28]. Overall, in this group, patients receiving 1L IO, either monotherapy or a combination with chemotherapy reached 23%. Considering the improved survival outcomes in the Post-1L IO group compared to the Pre-1L IO group, it is interesting to note that the small proportion of patients receiving 1L IO influenced the outcomes of the entire group. 

In the future, studies with an observational period following this analysis (after June 2020) are expected to include more patients treated with IO plus chemotherapy, and the clinical outcomes may differ to those reported here. Despite the bias of second- or later-line ICIs regarding the Pre-1L IO cohort (31.8% and 32.3% received second- and third-line ICIs, respectively), Post-1L IO patients showed better survival outcomes. This suggests that receiving ICIs as a first-line, rather than second- or later line, of treatment may be more favourable. 

Our study has several limitations. First, as seen in other retrospective observational studies, some clinical information of interest was not available. There was a relatively large proportion of patients with unknown EGFR, ALK and ROS1 status in both cohorts and the tumour PD-L1 expression was unknown for many patients in the Post-1L IO cohort. A small number of patients known to be living were excluded due to the lack of a signed informed consent form (17/895), whose exclusion may downwardly bias OS estimates. Moreover, the study dataset did not include information regarding comorbidities, concurrent medications and data on toxicity that would have helped to complete the characterization of the analysed [29,30]. The relatively small sample size, combined with the monocentric nature of the study, could limit the external validity of findings.

However, this study provides a more comprehensive picture of real-world population and documents the treatment pattern and survival outcomes of metastatic NSCLC patients without targetable mutations prior to and after the introduction in the Emilia-Romagna region of pembrolizumab as a 1L therapy. 

## 5. Conclusions 

This analysis provides insights into the current treatment use and outcomes of metastatic NSCLC patients in a real-world setting prior to and after the availability of immune regimens in 1L. Our findings provide evidence of improvements in overall survival associated with the introduction of 1L immunotherapy in real-world patients with metastatic NSCLC. In conclusion, 1L immunotherapy improved survival outcomes in patients with metastatic NSCLC and is now routinely used as a monotherapy or in combination with chemotherapy. 

## Figures and Tables

**Figure 1 cancers-14-04481-f001:**
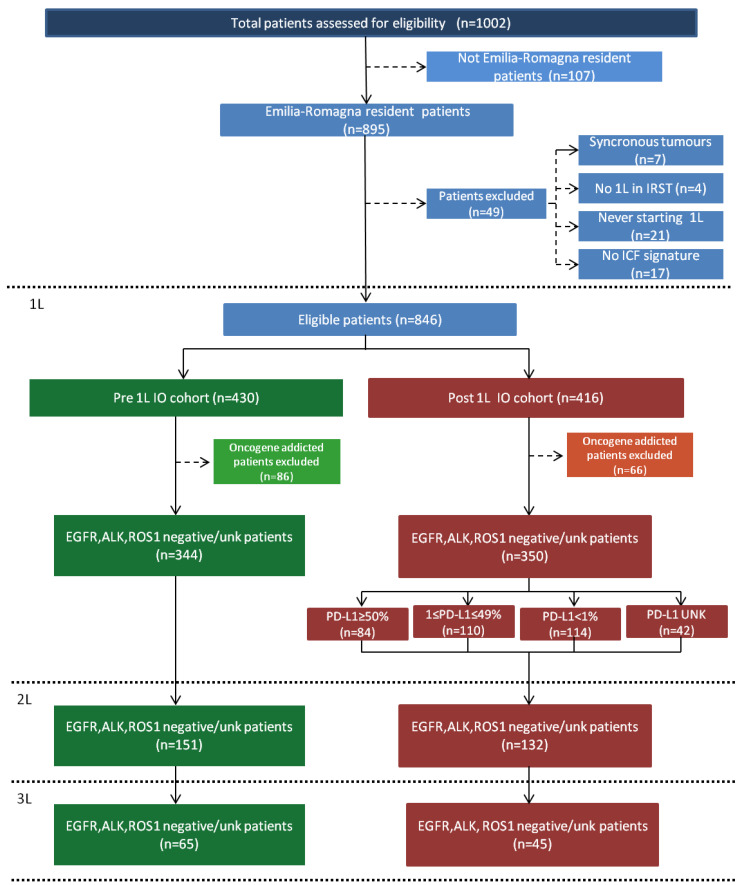
Patient flow chart. A total of 1002 NSCLC patients were assessed for eligibility. We excluded non-residents of the Emilia-Romagna region (*n* = 107). Other exclusions were made for patients with synchronous tumours (*n* = 7) patients who did not sign ICF (*n* = 17), patients who died before starting 1L treatment (*n* = 21) or received the 1L treatment in a different site to IRST (*n* = 4). Patients who met eligibility criteria (*n* = 846) were grouped into Pre- (*n* = 430) and Post-1L IO cohort (*n* = 416). A total of 86 and 66 patients were further excluded from Pre- and Post-1L IO, respectively, due to the oncogene-addicted tumor. A total of 694 patients were included in this analysis, grouped into Pre- (*n* = 344) and Post-1L IO (350).

**Figure 2 cancers-14-04481-f002:**
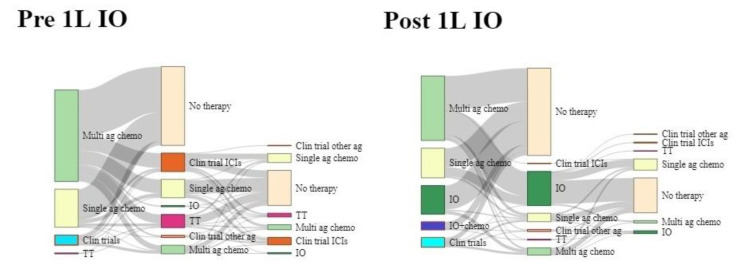
Treatment patterns across lines of therapy for Pre- and Post-1L IO cohorts. Each Sankey node represents a line of therapy, from 1L to 3L. The width of each group in each node was proportional to the number of patients. Possible reasons for not receiving 2L and 3L treatment include death, still receiving 1L or 2L treatment and patient preference. Multi ag chemo = multi-agents chemotherapy; single ag chemo = single-agent chemotherapy; clin trial = clinical Trials; TT = targeted therapy; no therapy = no further therapy, clin trial ICIs = clinical trial with ICIs agent; IO = immuno-oncology; clin trial other ag = clinical trial with other agents different from IO; IO + chemo = immuno-oncology+ chemotherapy.

**Figure 3 cancers-14-04481-f003:**
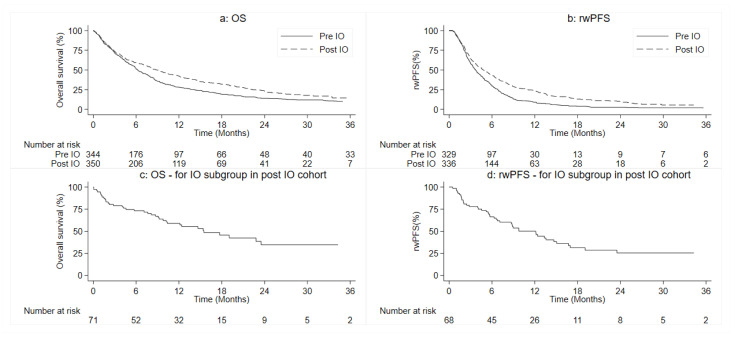
Estimation of probability of (**a**) OS and (**b**) rwPFS in 1L through Kaplan-Meier curves in the entire Pre-1L and Post-1L IO cohorts, respectively. Figures represent (**c**) OS and (**d**) rwPFS associated with the Post-1L IO subgroup with ICI as a first-line treatment.

**Table 1 cancers-14-04481-t001:** Baseline demographic and clinical characteristics of patients by Pre- and Post-1L IO groups.

Characteristics	Pre-1L IO *n* = 344 (%)	Post-1L IO *n* = 350 (%)
**IIIBrp */IV stage**		
IIIBrp	14(4.1)	6(1.7)
IV	330(95.9)	346(98.3)
**Age at IIIBrp /IV stage diagnosis**		
<70 years	190 (55.2)	152 (43.4)
70–74 years	78 (22.7)	81 (23.1)
75–79 years	46 (13.4)	82 (23.5)
80–84 years	26 (7.6)	25 (7.1)
≥85 years	4 (1.1)	10 (2.9)
**Gender**		
Female	118 (34.3)	112 (32.0)
Male	226 (65.7)	238 (68.0)
**Race**		
White	341 (99.1)	350 (100.0)
Others	3 (0.9)	0 (0.0)
**Smoking history**		
Never	13 (4.7)	23 (8.2)
Ever	266 (95.3)	258 (91.8)
Unknown	65	69
**Year smoked**		
≤20 years	19 (8.8)	18 (11.2)
>20 years	196 (91.2)	143 (88.8)
Unknown	129	189
**Packs/year**		
≤20 packs/years	26(13.0)	24(15.4)
˃20 pack/years	174(87.0)	132(84.6)
Unknown	144	194
**ECOG PS at IIIBrp/IV stage diagnosis**		
0	62(18.9)	55(16.9)
1	217 (63.1)	212(65.0)
≥2	49 (14.9)	59 (18.1)
Unknown	16	24
**Histology**		
Squamous cell	57 (16.8)	70 (20.5)
Non-squamous cell	263 (77.4)	267 (77.8)
*Adenocarcinoma*	*259 (76.2)*	*267 (77.8)*
*Large cell carcinoma*	*4 (1.2)*	*0 (0.0)*
Other	20 (5.8)	6 (1.7)
Unknown	4	7
**Unknown biomarker status**		
EGFR unknown	85 (24.7)	99 (28.3)
ALK unknown	142 (41.3)	109 (31.1)
ROS-1 unknown	275 (79.9)	122 (34.9)
**Location of metastasis**		
Bone	114 (33.1)	99 (28.3)
Lymph nodes	73 (21.2)	95 (27.1)
Brain	44 (12.8)	55 (15.7)
Liver	33 (9.6)	32 (9.1)
Pleura	48 (14.0)	56 (16.0)
Contralateral lung	121 (35.2)	122 (34.9)
Other	92 (26.7)	55 (15.7)
Missing/Unknown	3 (0.9)	6 (1.7)

* IIIBrp = stage IIIB with rapidly progressive who experienced disease progression to stage IV within 6 months of the first anticancer treatment.

**Table 2 cancers-14-04481-t002:** First-line treatments administrated by Pre- and Post-1L IO cohorts. Post-1L IO cohort was further grouped based on PD-L1 TPS value (TPS ≥50% and TPS <50%).

	Pre-1L IO	Post-1L IO	Post-1L IO	
First-Line Therapies	*n* = 344 (%)	*n* = 350 (%)	TPS ≥50% *n* = 84 (%)	TPS <50% *n* = 266 (%)
**Multi-agents chemotherapy**	**224 (65.1)**	**161 (46.0)**	**6 (7.1)**	**155 (58.2)**
*Carboplatin + Gemcitabine*	*92 (26.7)*	*99 (28.3)*	*1 (1.1)*	*98 (36.8)*
*Pemetrexed +/− Platin*	*123 (35.8)*	*51(14.6)*	*2 (2.4)*	*49 (18.4)*
*Other combinations*	*9 (2.6)*	*11 (3.1)*	*3 (3.6)*	*8 (3.0)*
**Single-agent chemotherapy**	**93 (27.0)**	**74 (21.1)**	**3 (3.6)**	**71 (26.7)**
*Gemcitabine*	*51 (14.8)*	*35 (10.0)*	*2 (2.4)*	*33 (12.4)*
*Vinorelbine*	*36 (10.5)*	*38 (10.9)*	*1 (1.2)*	*37 (13.9)*
*Other agents*	*6 (1.7)*	*1 (0.3)*	*-*	*1 (0.4)*
**Targeted therapy**	**2 (0.6)**	**0 (0.0)**	**0 (0.0)**	**0 (0.0)**
**PD-1/PD-L1 inhibitor single agent**	**-**	**71 (20.3)**	**71 (84.5)**	**0 (0.0)**
*Pembrolizumab*	**-**	*71(20.3)*	*71 (84.5)*	*0 (0.0)*
**PD-1/PDL1 inhibitor + chemotherapy**	**-**	**20 (5.7)**	**0 (0.0)**	**20 (7.5)**
**Clinical Trials**	**25 (7.3)**	**24 (6.9)**	**4 (4.8)**	**20 (7.5)**

## Data Availability

The data presented in this study are available on a reasonable request from the corresponding author.

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
