# Peer review of "Real-World Outcomes and Treatments Patterns Prior and after the Introduction of First-Line Immunotherapy for the Treatment of Metastatic Non-Small Cell Lung Cancer"

_cancers, 2022, doi:10.3390/cancers14184481_

Round 1
Reviewer 1 Report
The electronically submitted manuscript “Real World Outcomes and Treatments Patterns Prior and After the Introduction of First-Line Immunotherapy for the Treatment of Metastatic Non-Small Cell Lung Cancer” by Danesi V et al, is an original paper aiming to describe the patients’ outcomes before and after the local regulatory approval of 1L pembrolizumab to complement the growing real-world literature on ICIs in the treatment of NSCLC patients. The authors performed a descriptive analysis using data from electronic health records and found that 1L immunotherapy has improved survival outcomes. This is an overall interesting study. Below I leave some minor comments so that the author could take into consideration, if they wish, for a revised version of their manuscript.
Consider changing “real world” to “real-world”. Also, I have the feeling that this term is repeated many times in the manuscript (for example, it is part of many variables).
Line 57: write what the abbreviation PD1/PDL1 stands for
Line 61: the same for AIFA
Lines 60-73: Is it possible to add references?
Line 125: write what the abbreviation TPS stands for
rwPFS: Is there any difference between PFS and rwPFS? Is the “real-world” necessary in this term? If not, consider removing it (for clarity) unless it is used in the literature. (This is just a doubt/suggestion to think about). rwTRT: the same. Real-world disease control rate: the same.
Figures and Tables should be self-explained. So, I suggest giving the meaning of the abbreviations in the legends.
Figure 2: Well-made figure and very informative. Just a question: is the program that you used for making this figure named in the methodology section? If not, consider adding it (only if necessary)
Lines 264-268: is there any special reason this paragraph should be in italics?
Discussion: Add a first paragraph in the discussion section highlighting the most important findings of your research and briefly comment on their meaning and applicability.
How did the authors deal with missing data?
Reviewer 2 Report
It is important to determine the tolerability of immunotherapy and therefore
data regarding toxicity should be included.
Please explain better what therapeutic schedules with platinum were most commonly used.
It is important to define the main comorbidities and concurrent medications to which patients underwent.
Why did you choose pack-years of 40 as a cutoff level since screening programs considered a level of 20 to be enough consider it a risk factor for lung cancer. It seems that pembrolizumab is more effective as first line, please clarify BETTER in the text the meaning of pre and post IO
Did you use log rank test to compare the survival curves?
I suggest to include the following references useful for discussion
-Crit Rev Oncol Hematol. 2019 Oct;142:26-34.
-Lung Cancer. 2022 Aug 6;172:65-74.
Reviewer 3 Report
This article provides insights into treatment use and outcomes of metastatic non-small cell lung cancer (NSCLC) patients in a real-world setting prior to and after the availability of immuno-oncology (IO) regimens in first-line (1L). The study suggests that receiving immunotherapy in the first line rather than in the second or later lines of treatment may be more favorable.
This article provides real-world data support for first-line applications of immunotherapy in NSCLC. I agree with the publication of this article.
Author Response
Thank you very much.